# Endometriosis: A Disease with Few Direct Treatment Options

**DOI:** 10.3390/molecules27134034

**Published:** 2022-06-23

**Authors:** Patricia Ribeiro de Carvalho França, Anna Carolina Pereira Lontra, Patricia Dias Fernandes

**Affiliations:** Laboratório de Farmacologia da Dor e da Inflamação, Instituto de Ciências Biomédicas, Universidade Federal do Rio de Janeiro, Rio de Janeiro 21941-902, Brazil; patriciaribeiro.ufrj@yahoo.com.br (P.R.d.C.F.); anna.lontra@gmail.com (A.C.P.L.)

**Keywords:** endometriosis, gynecological disease, inflammation, drug therapy

## Abstract

Endometriosis is a gynecological condition characterized by the growth of endometrium-like tissues inside and outside the pelvic cavity. The evolution of the disease can lead to infertility in addition to high treatment costs. Currently, available medications are only effective in treating endometriosis-related pain; however, it is not a targeted treatment. The objective of this work is to review the characteristics of the disease, the diagnostic means and treatments available, as well as to discuss new therapeutic options.

## 1. Introduction

### 1.1. Endometrial Physiology and Menstrual Cycle

The uterus, a structure of mesodermal origin, is a hollow muscular organ with thick walls and unique anatomical and histological characteristics. It has two uterine tubes that attach to the uterus, creating a path for the passage of ovum from the ovaries to the uterus, for later fertilization and implantation in the endometrium [1]. The uterine wall consists of several layers: the perimetrium, a serous lining; an outermost layer that comprises the peritoneum and is supported by surrounding layers of connective tissue; the middle layer of the uterus known as the myometrium, one of the most dynamic layers of the uterus, suffering great distensions to allow the accommodation of various arteries and nerves, especially during pregnancy; and the endometrium, the innermost lining of the myometrium and the most active layer, as it undergoes different changes during the menstrual cycle [2].

The uterine endometrium has the function of preparing the uterus for the implantation of the embryo, maintaining the pregnancy and, if this does not occur, starting the menstrual cycle, with the shedding of the endometrium. Cell types found in the endometrium include stromal, epithelial, vascular, and immune cells [3].

### 1.2. Endometriosis

Currently, 190 million women worldwide present with endometriosis [4]. Endometriosis is a chronic, hormone-dependent inflammatory disease defined by the presence of foci of endometrial tissue outside the uterine cavity, which can be found mostly in the pelvic cavity, but can also be found in the ovaries, fallopian tubes, sigmoid colon, appendix, upper abdomen, among others [5]. In the general population, rates of the disease are difficult to quantify due to difficulties in a definitive diagnosis and asymptomatic cases, thus aggravating the condition of endometriosis. Therefore, estimates vary widely between different population samples and modes of diagnosis, all influenced by the presentation of symptoms and access to care [6]. Despite this limitation, the prevalence of endometriosis is greater than 10% in women in the reproductive period [7]. It is observed that 90% suffer from dysmenorrhea, 76% from dyspareunia, 77% from chronic pelvic pain, 66% from dyschesia, and 15% from hematochezia [8]. One of the most common complications in endometriosis is infertility, a symptom that occurs in 50% of women diagnosed with the disease (Figure 1). These symptoms compromise women’s quality of life, especially in relation to noncompliance with activities of daily living, personal relationships, and low productivity at work [9]. In addition, many studies report the correlation between anxiety and depression with pain symptoms and loss of fertility, leading to a rate of almost 87% of women with endometriosis developing some type of psychiatric disorder [10].

Endometriosis impacts different aspects of life, and one of them is the economic aspect. A large multicentre study carried out between Europe, UK, and the USA showed that the total cost per woman with endometriosis per year was approximately EUR 9579.00, with most of the costs (EUR 6298.00) caused by the absence or reduction of effectiveness at work resulting from the symptoms of the disease [11,12]. Medical care costs (EUR 3281.00) were mainly due to surgeries (29%), monitoring exams (19%), hospitalization (18%), and medical visits (16%) [13].

### 1.3. The Origin of Endometriosis

One of the first questions about the pathogenesis of endometriosis was the origin of glands and stroma such as the eutopic endometrium outside the uterine cavity, characteristics which constitute the disease. Several hypotheses have been proposed since 1870; however, none fully explain the pathogenesis of endometriosis [14]. Among the existing theories, the following stand out: the theory of coelomic metaplasia; the theory of Müller’s embryonic development; the theory of lymphatic and vascular metaplasia; the theory of implantation of endometrial stem cells; and the theory of retrograde menstruation (Figure 2) [15,16].

#### 1.3.1. Coelomic Metaplasia Theory

In 1924, Robert Meyer first suggested that endometriosis could originate from coelomic metaplasia [17,18]. This theory supports the hypothesis that multipotent mesenchymal stem cells, derived from bone marrow or from a niche within the endometrium itself, undergo a reprogramming process in which they can differentiate into endometrial and stromal epithelial cells at ectopic sites [17]. It is believed that endocrine disruptors and hormonal or immunological factors may play an important role in this transformation [19]. This theory is widely used to explain the origin of rare cases in which endometriosis occurs in sites outside the pelvic cavity, including abdominal lymph nodes, lungs, brain, kidneys, and cases of Müllerian agenesis (i.e., the congenital malformation in which the Müller’s ducts do not develop) [20,21] including, in very rare cases, where endometriosis has been observed in men [16,20].

#### 1.3.2. Müllerian Embryonic Remnant Abnormalities Theory

Müller’s ducts are primordial embryological structures that, during fetal life, develop to form the reproductive system. With respect to the female reproductive system (uterus, fallopian tubes, and upper vagina), these ducts consist of surface epithelium and mesenchyme from the underlying urogenital crest, capable of differentiating into endometrial glands and stroma [22]. The theory of Müller’s embryonic development, or Müllerians, postulates that, during organogenesis, Müller’s duct cells undergo a process of disordered differentiation and proliferation [23], causing them to spread to localized sites outside the expected area of Müller’s duct development [14]. In this way, residual cells arising from embryological migration of the Müller duct maintain the ability to develop into endometriotic lesions under the influence of the hormone estrogen, starting at puberty or perhaps in response to estrogen mimetics [24].

#### 1.3.3. Lymphatic and Vascular Metastasis Theory

This hypothesis states that endometrial cells and tissue fragments are transported from the uterine cavity through blood or lymph vessels to colonize distant ectopic sites such as the lung, diaphragm, abdominal wall, or brain. This hypothesis best describes the rare occurrence of extrapelvic endometriosis in women and is supported by evidence of endometrial cell emboli in sentinel lymph nodes [25,26].

#### 1.3.4. Retrograde Menstruation Theory

Through clinical and anatomical observations, Sampson (1927) proposed the theory of retrograde menstruation, which is considered to be the most accepted theory that explains most types of endometrioses [27]. This theory postulates that, during menstruation, blood and fragments of viable endometrial tissue flow back into the peritoneal cavity through the fallopian tubes, where they subsequently implant into the peritoneal tissue and/or the pelvic organs, which is the first step in the development of endometriotic lesions [15,28,29]. Evidence for this theory was demonstrated by the observation of menstrual residues in the peritoneal cavity during menstruation [28]. However, this process is not considered pathological, as it occurs in 76–90% of healthy women, but not all of them develop the disease. Thus, other factors favor the survival, invasion, and vascularization of endometriotic lesions [30].

#### 1.3.5. Endometrial Stem Cell Implantation Theory

The theory of stem cell implantation has gained considerable attention in recent years. The two main variants of the theory are based on tissue derived from stem cells, where they are believed to come from the uterine endometrium or bone marrow [31]. Regardless of where stem cells originate, hormones and other factors in the tissue microenvironment contribute to adhesion, invasion, inflammation, angiogenesis, and evasion of the immunosurveillance necessary for the establishment of endometriosis [32]. The core of this theory is the dissemination of these cells by different mechanisms such as retrograde menstruation, lymphatic and vascular dissemination, direct migration, or a combination of all, to the implantation site. In this location, influenced by factors not yet fully elucidated, these multipotent cells begin their proliferation process, often dependent on hormonal cycles, especially estrogen [33,34]. Thus, the differential of this theory is that it not only fits the model of retrograde menstruation, but also explains the pathogenesis of deep infiltrating endometriosis and endometriosis outside the abdominal cavity [35].

### 1.4. Classification

The challenge in classifying endometriosis is to correlate the progression of the disease with infertility and pain, which are the most relevant clinical features. This is undoubtedly the greatest difficulty encountered by physicians and the main complaints of women affected by the disease [36,37]. In 1979, the American Fertility Society (AFS) first proposed a classification system. The AFS system was flexible, quantitative, and the analysis was assigned by scores. Over the years, important modifications have been made and the system has been revised (rAFS, revised American Fertility Society). In 1996, it was renamed the American Society for Reproductive Medicine (rASRM, revised American Society for Reproductive Medicine) [38]. This system relates the stages of endometriosis with arbitrary cumulative points, which are given according to the surgical assessment of the size, location, severity of endometriotic lesions, and the occurrence of adhesions. Thus, women with endometriosis are divided into four stages: I (1–5 points), II (6–15 points), III (16–40 points), and IV (>40 points). The highest points in this system consider the presence of ovarian endometrioma greater than 3 cm (20 points on each side), complete blockage of the cul-de-sac (40 points), and the presence of ovarian or tubal adhesions (16 points) [9,36].

The main advantages of the rASRM system include the fact that it is widely used and accepted around the world, is easy to classify, and is easy for patients to understand. However, there are some limitations/criticisms with the use of this system, such as arbitrary scores not relating the symptoms of pain and infertility and not including deep lesions, which affect different sites such as uterosacral ligament, retroperitoneal structures, bladder, vagina, and intestine [39]. In 2005, the Endometriosis Scientific Foundation published the Enzian classification in order to complement the rASRM scores, especially with respect to deep lesions. This classification considers three axes or compartments (a, b, and c) and numbers, which correlate the locations and size (>5 mm) of the lesions, respectively [40,41,42]. Thus, the Enzian score is a descriptive morphological classification that allows multiple nominations when there are several foci of the lesion in different organs. However, the two classifications cited above do not consider the main symptoms of endometriosis [38].

Based on this context, in 2014, the World Endometriosis Society (WES) proposed a consensus for the classification of endometriosis, which included representatives of medical and nonmedical organizations (national and international), where they formulated a declaration proposing a classification system, until a new and improved system is developed. This includes the rASRM classification, the Enzian classification, and the Endometriosis Fertility Index (EFI), which in turn aims to provide a prognosis on the fertility status of women with endometriosis [37].

The EFI encompasses historical factors (age, duration of infertility, and pregnancy history) and surgical factors (rASRM scores, postsurgical procedure scores, and the sum of both), which when added together result in an estimated percentage of the pregnancy rate. The EFI ranges from 0 to 10 points, with 0 representing the worst prognosis and 10 the best prognosis. This system can be used to decide what type of treatment patients can opt for, for how long and at what cost, before considering assisted reproductive technologies (ARTs) after undergoing endometriosis surgery [43,44]. In 2007, the American Association of Gynecological Laparoscopists (AAGL) started a project to develop a new classification of endometriosis. This system was based on relating pain, infertility, and surgical difficulties. At a meeting in Las Vegas, the AAGL special interest group presented promising preliminary results from this correlation. However, this classification has not yet been fully validated and published, although more than ten years have passed since the classification was proposed, thus showing the need for further investigation and discussion of the AAGL classification [42].

Endometriosis can also be classified into three different phenotypes based on their histopathology and anatomical locations: Superficial Endometriosis (SE), Deep Infiltrating Endometriosis (DIE), and Ovarian endometriotic Cysts (EC) (known as endometriomas or chocolate cysts). Superficial endometriosis usually appears on the surface or subserous soft tissue of the peritoneum or visceral organs and is the mildest form of the disease. ECs are found in the ovary, usually forming a large cystic structure (clinically interpreted as an adnexal mass). DIE, on the other hand, is a more severe phenotype, involving subperitoneal lesions that extend deeper, >5 mm, into the muscular layer of the intestine, bladder wall, diaphragm, or other organs [16].

## 2. Etiopathogenesis of Endometriosis: Changes in the Main Pathways

The development of endometriotic lesions involves the interaction of several systems, so endometriosis is considered a multifactorial disease [6]. The pathophysiology of the disease is closely linked to genetic factors; hormonal factors such as progesterone resistance or estrogen dependence; inflammatory processes such as an increase in inflammatory mediators, angiogenesis, oxidative stress, or immunological factors, which are also involved in the development of a lesion [45]. Each of these factors will be discussed in detail below.

### 2.1. Genetic and Epigenetic Changes

The first genetic studies in patients with a proven diagnosis of endometriosis appeared in the 1980s [46]. Over the years, several authors have shown the relationship between endometriosis and heredity through family aggregation studies, which in turn concerns the accumulation of specific characteristics found in a given family and cannot be attributed to coincidental events [47]. Simpson et al. [48] report that when a mother has endometriosis, the probability of her daughters developing the disease is 8%, and when a sister is affected, the probability is 6%. In the control population, the risk of daughters developing the disease is less than or equal to 1% in both situations. In addition, there is a correlation between patients who have a positive family history (with several cases in the family) with greater chances of presenting severe manifestations of the disease [49].

The identification of genetic variants that influence the likelihood of endometriosis development may be a crucial factor in the pathogenesis of the disease [17]. In view of this, Genome-Wide Association Studies (GWAS) have provided new tools to identify genetic variations (Single Nucleotide Polymorphisms, SNPs) in patients with endometriosis. These studies showed several SNPs at different loci, among which the following stand out: VETZ, a gene involved in cell growth, migration, and adhesion; CDKN2B-AS1, a gene that controls specific tumor suppressors, where its inactivation has been correlated with the development of endometriosis and endometrial cancer; WNT-4, a gene involved in the development of the female reproductive system, essential in the formation of Müller’s duct; GREB1, involved in estrogen regulation; and ID4, found in cosuppressor genes involved in human ovarian tumors. Some of these loci have a strong relationship with advanced cases of endometriosis, classified by ASRM into stages III and IV, showing a correlation between SNPs with moderate to severe disease development [14,15,49,50].

The relationship between endometriosis and ovarian cancer risk has prompted several studies to conduct somatic mutational analyses. Through exome sequencing, it was observed that 79% of the lesions (EC and/or DIE) had mutations, and 26% of these were found in the ARID1A, PIK3CA, KRAS, and PPP2R1A genes, where PIK3CA and KRAS are genes frequently mutated in ovarian cancer. These findings may partially explain the aggressive nature of DIE compared to SE lesions. Furthermore, these mutations may play critical roles in the implantation and establishment of both ovarian and extraovarian endometriotic lesions, although they may not become malignant [50,51,52].

Epigenetic modifications are reversible changes in a cell’s DNA or histones that regulate gene expression without altering the DNA sequence. Two of the most characterized epigenetic (single genes) or epigenomic (entire genome) modifications are DNA methylation (cytosine residues) and histone modification (methylation or acetylation of specific histones in chromatin) [50]. Numerous evidence show that endometriosis is an epigenetic disease, the first of which shows that the promoter of the HOXA10 gene is hypermethylated in the endometrium of women with endometriosis when compared to healthy women without the disease. This gene is expressed in the eutopic endometrium and is dramatically increased during the secretory phase of the menstrual cycle, corresponding to the time of implantation and the increase in circulating progesterone. Hypermethylation is usually associated with gene silencing, causing changes in transcription and reduced expression. The reduction of HOXA10 in women with endometriosis is related to defects in uterine receptivity, which could be an explanation for the low fertility rate of these women [53]. In addition, hypermethylation of the HOXA10 gene results in a reduction in the expression of E-cadherin, an intercellular adhesion molecule, and therefore, may cause a rupture of intercellular junctions with loss of polarization and epithelial morphology, favoring the process of cell invasion [46].

The promoter region of the progesterone receptor-B (PR-B) is also hypermethylated, and therefore leads to a reduction in the expression of PR-B; this scenario is partially responsible for resistance to progesterone, one of the main features of endometriosis [54]. On the other hand, some genes may be hypomethylated, resulting in increased expression of the same. The main genes or promoter regions where hypomethylation occurs are the estrogen receptor-β (ER-β), steroidogenic factor (SF-1, Steroidogenic Factor-1), and aromatase. SF-1 is a transcription factor that activates several genes for estrogen biosynthesis, which in turn is not detected in eutopic endometrial stromal cells because the SF-1 promoter is usually hypermethylated in endometrial cells. Ectopically, the SF-1 promoter was reported to be hypomethylated, explaining its overexpression and consequent increase in estrogen. The same is true for the aromatase gene and the ER-β promoter, where overexpression in ectopic endometrial cells also leads to an increase in estrogen and its receptor, respectively [55].

### 2.2. Dysfunction of the Innate Immune System and Its Relationship with the Development of Peritoneal Endometriotic Lesions

Part of the etiopathogenesis of endometriosis is related to a dysfunction of the immune system. This disease is marked by numerous alterations, among which are the peritoneal infiltration of immune cells, macrophage activation, alteration of the cytotoxicity of natural killer cells (NK), as well as the excessive production of proinflammatory cytokines, factors that directly influence the adhesion, implantation, and survival of ectopic endometrial cells [56,57].

In women with endometriosis, there is a significant infiltration of these cells in the peritoneal fluid, consequently, neutrophil chemotactic factors such as interleukin-8 (IL-8), granulocyte colony stimulating factor (G-CSF), chemokine ligands 1, 2, and 3 (CXCL-1, CXCL-2 and CXCL-3, CXC Motif Chemokine Ligand 1, 2 and 3), among others, are also increased in the peritoneal fluid, leading to positive feedback for the recruitment of this cell type. In addition, these cells release several mediators, such as tumor necrosis factor (TNF-α), the interleukins (IL) IL-6, IL-8, IL-1β, IL-10, and vascular endothelial growth factor (VEGF) [58,59]. In a murine model, it was demonstrated that neutrophil depletion using the antigranulocyte receptor-1 antibody (anti-Gr-1) reduced the formation of endometrial lesions in the early stage of endometrial development disease, thus suggesting that neutrophils are essential for the initial formation of endometriosis [60]. In the peritoneal fluid, macrophages are the most prevalent immune cells, and have two main functions of phagocytosis, removing red blood cells, damaged tissue fragments and cellular debris (from retrograde menstruation), and the production of inflammatory mediators. Many studies have shown a significant increase in macrophages in the peritoneal fluid and in the eutopic endometrium of women with endometriosis; however, there was also a low expression of CD36 and a reduction in the activation of matrix metalloproteinases, components that regulate the phagocytic activity of macrophages, where both mechanisms are suppressed by the overexpression of prostaglandin E2 (PGE2). In addition, the number of endometrial cells disseminated in the peritoneal cavity may be greater than the ability of macrophages to remove them, causing an intense and very short menstrual flow, thus, facilitating the process of implantation of ectopic endometrial cells in the peritoneal cavity [28,61].

Another important factor to consider is the plastic nature of these cells and their ability to adopt a classically activated macrophage (M1) or alternatively activated macrophage (M2) phenotype in response to environmental stimuli [61]. Macrophages with an M1 phenotype produce proinflammatory cytokines and chemokines and are specialized in eliminating microorganisms and defective cells, whereas macrophages with a M2 phenotype modulate the adaptive immune response, promote angiogenesis, tissue repair, and secrete both anti-inflammatory cytokines and anti-inflammatory growth factors [62]. In the peritoneal fluid of women with endometriosis, there is an imbalance between M1 and M2 macrophages; although most of the literature remains controversial, the majority of studies show an increase in macrophage polarization for the M2 phenotype, reported mainly in models of endometriosis in mice. In contrast, macrophages with an M1 phenotype are found in large numbers in the eutopic endometrium of women affected by the disease [14,61,62,63,64]. Ectopic endometrial cells can induce the expression of certain enzymes and proteins that favor macrophage polarization to the M2 phenotype, such as indoleamine 2,3-dioxygenase-1 (IDO1), which promotes an inflammatory response and subsequently initiates macrophage polarization to the M2 phenotype. All the factors exposed above will favor the adhesion and proliferation of ectopic endometrial cells, as well as the development of the disease [56,65].

Like macrophages, NK cells are key components of the innate immune system. They are characterized by the ability to mediate cytotoxic activity, releasing granzyme, perforin and cytokines, such as interferon-γ (IFN-γ), from their cytolytic granules to destroy invading cells. For this reason, they are considered the first line of defense against viruses, viral infections, and growth of tumor cells [65,66]. However, in the peritoneal fluid of women with endometriosis, NK cells exhibit a reduction in cytotoxicity; this fact is due to the activation or inhibition of surface receptors found in these cells, directly impacting their function. Some studies have shown that the natural receptor for cytotoxicity, NKp46, and the cell surface marker for cytotoxicity, CK107a, are significantly reduced in the peritoneal fluid of patients with endometriosis. In contrast, the NKG2A receptor, an inhibitory cytotoxic receptor, is overexpressed in peritoneal NK cells [57,67,68]. The increase in cytokines in the peritoneal fluid, as well as those secreted by the ectopic endometrial tissue itself, also interfere with the functionality of NK cells. These cytokines are the platelet-derived transforming growth factor β (TGF-β), which mediates the negative modulation of NKG2D expression (receptor that triggers cytolytic activity and cytokine production); IFN-γ that loses its function of stimulating apoptosis of ectopic endometrial cells; IL-6, which is responsible for inhibiting the cytotoxicity of NK cells and negatively regulating the expression of granzyme and perforin; and IL-15, which is secreted by ectopic endometrial tissue cells and directly interferes with the reduction of the cytotoxic activity of NK cells through the reduction of granzyme B, IFN-γ, and NKG2D and NKp44 receptors (natural cytotoxicity receptor), thus contributing to the immune escape of ectopic endometrial cells into the peritoneal cavity, favoring the implantation, proliferation, and progression of endometriosis [56,57,69].

It is also important to mention the myeloid dendritic cells. These cells have heterogeneous populations of antigen-presenting cells highly involved in the initiation and modulation of the immune response; therefore, changes in these cells can affect the progression and pathogenesis of endometriosis [70]. Depending on the origin and secreted molecules, dendritic cells are classified into plasmacytoid (lymphoid origin) and myeloid (hematopoietic origin). The latter is found in eutopic endometrium, ectopic endometrial lesions, and in the peritoneal fluid of women affected by the disease [66]. Recent studies report the expression of mannose receptors in myeloid dendritic cells, which then phagocytize ectopic endometrial cell residues. However, this action contributes to the inflammatory profile observed in patients with endometriosis through the secretion of IL-6 and IL-1β by these cells [28]. Findings by Hey-Cunningham and collaborators [71] showed that changes in dendritic cell populations, with a significant increase in the density of immature myeloid cells and a reduction of mature myeloid cells, was observed in women with endometriosis compared to women without the disease. In murine models, immature cells exhibit high proangiogenic activity (formation of new vessels from pre-existing ones) similar to the process of tumor angiogenesis, favoring the growth and establishment of ectopic endometrial lesions [72]. In addition, plasmacytoid dendritic cells secrete IL-10 which promote the growth of lesions as it stimulates angiogenesis through VEGF-dependent or independent pathways [73]. However, several evidences indicate that not only the dysfunction of immune cells and the increase in the secretion of several cytokines lead to adhesion and implantation of the ectopic tissue, but also, the increase in molecules that modulate the cell–cell and cell–matrix interactions expressed by the ectopic endometrial lesions [14,74].

The increased expression of E-cadherins and P-cadherins in the peritoneal mesothelium of women with endometriosis contributes to the adhesion of ectopic endometrial cells, while the increased expression of N-cadherins is related to the loss of junctions between cells, peritoneal membranes, reorganization of the cytoskeleton, and expression of vimentin and actin, favoring the migration and invasion of ectopic endometrial cells [75]. In addition, mesothelium invasion can also be facilitated by matrix metalloproteinases (MMPs), a group of proteins that degrade and remodel the extracellular matrix, forming space between cells, thus promoting migration [74]. MMPs are present in the peritoneal fluid and in endometriotic lesions, and these can be regulated by cytokines and steroid hormones [56]. Furthermore, the imbalance between MMPs and tissue inhibitors of metalloproteinases (TIMPs) also contributes to the remodeling of the extracellular matrix, further promoting the invasion, migration, and development of endometriotic lesions [75].

### 2.3. Interaction between Steroid Hormones and Inflammation in Endometriosis

Deregulation of steroid hormone signaling is common in several uterine pathologies, such as endometriosis, infertility, leiomyoma, endometrial cancer, among others. Steroid hormones play a key role in the development and establishment of ectopic endometrial lesions, which occur through resistance to progesterone and a significant increase in estrogen, mediated by an inflammatory environment. Resistance to progesterone occurs when the tissue does not respond adequately to this hormone, caused by a failure in the activation of the receptor or in the transcription of the target gene. Several studies have shown that progesterone resistance occurs both in endometriotic lesions and in the eutopic endometrium of women with endometriosis. Loss of responsiveness to this hormone can have serious consequences in both cases, as progesterone signaling is necessary to counteract estrogen-induced proliferation and promote decidualization. Thus, progesterone resistance can lead to increased lesion growth and a nonreceptive endometrium [50,76].

Chronic inflammation is one of the many factors that can influence the expression of the Nuclear factor-κB (NF-κB). It is an important transcription factor activated by several cytokines that are increased in the peritoneal fluid of women with endometriosis. This, in turn, acts directly on the expression of the progesterone receptor, through an antagonistic effect [77]. The cytokines TNF-α and IL-1β directly decrease the levels of both isoforms of the progesterone receptor (PR-A and PR-B), probably through epigenetic modifications (i.e., hypermethylation of the progesterone receptor). However, proinflammatory cytokines can also interfere with the function of these receptors through direct competition with them or by connecting the receptor with transcription factors, thus influencing the expression of key genes [78]. Thus, progesterone resistance leads to the deregulation of its genes, for example TOB-1, a cell cycle inhibitor; glycodelin, an immunomodulatory protein and marker of differentiated endometrial function; and 17β-hydroxysteroid dehydrogenase type 2 (17β-HSD2), responsible for converting estradiol to estrone (a less active form of estrogen) [17]. The loss of the function of this last enzyme plays a crucial role in the formation of ectopic implants, as it contributes to the increase of estrogen at the site of implantation of the cysts [17,24,79]. In endometriotic stromal cells, an increase in the expression of steroidogenic factor-1 (SF-1) was observed, a transcription factor that favors the expression of the aromatase P450 gene [17]. This enzyme is responsible for converting androgen into estrogen and is expressed both in ectopic lesions and in eutopic endometrial tissue of women with endometriosis [80,81]. The increased expression of aromatase P450, together with the loss of 17β-HSD2 function, leads to a significant accumulation of estrogen; consequently, there is also an increase in the expression of its receptor, the ER-α and ER-β isoforms. Several evidences show an increase in the ER-β/ER-α ratio in both eutopic endometrium and ectopic endometrial tissue. The imbalance between the proportions of the isoforms is mainly due to epigenetic changes [16,74,79].

High local concentrations of estrogen promote mitogenic activity by activating a series of genes related to cell proliferation (GREB1, MYC, and CCND1); the survival of endometriotic cells, when interacting with the apoptosis signal-regulating kinase-1 (ASK-1), reducing its activity and, therefore, inhibiting the apoptosis induced by TNF-α, thus, leading to the evasion of these cells from the immune surveillance system; the proliferation and adhesion of endometriotic cells, when related to some components of the inflammasome, NLRP3 and caspase 1, through the secretion of the cytokine IL-1β; the resistance of the progesterone receptor, decreasing its expression in endometriotic lesions; and the epithelium–mesenchymal transition [82]. In addition, the increased production of estrogen in endometriotic lesions and in the eutopic endometrium leads to the induction of the production of the enzyme cyclooxygenase-2 (COX-2), which subsequently also increases the production of the mediator prostaglandin E2 (PGE2). In turn, this further stimulates aromatase activity, resulting in a positive feedback loop, thus increasing the concentration of local estrogen [76,79,80]. PGE2 stimulates the expression of all steroidogenic genes in endometrial stromal cells, thus leading to increased levels of this hormone. In addition, this mediator also plays an important role in the survival and growth of endometriotic lesions. Via protein kinase A (PKA), PGE2 regulates endometriotic cell proliferation through the phosphorylation and nuclear translocation mechanism of the Ras-like and Estrogen-Regulated Growth (RERG) gene, as well as induces the expression of the glucocorticoid-regulated kinase-1 (SGK1) that contributes to survival of endometriotic cells through the inhibition of apoptotic mechanisms [79,83]. The COX-2/PGE2 complex also induces VEGF expression and promotes angiogenesis and lesion growth [84]. Arosh and collaborators [85] demonstrated that inhibition of specific PGE2 receptor subtypes (EP2 and EP4) suppresses survival, invasion, biosynthesis, and signaling of PGE2 and estrogen; stimulates production of proinflammatory cytokines; and increases proteins related to progesterone signaling in epithelial and stromal cells of endometriotic lesions.

The estrogenic microenvironment can activate peritoneal macrophages as well as induce the translocation of NF-κB to the nucleus, both with a consequent increase in pro-inflammatory cytokines locally and systemically. Furthermore, these mechanisms also induce VEGF expression, cell cycle activation, and activation of antiapoptotic genes [14,86]. Several cytokines are elevated in the peritoneal fluid and in the serum of patients with endometriosis; however, the cytokines IL-1β, TNF-α, IL-6, and IL-10 play a fundamental role in the pathogenesis of endometriosis [87]. The cytokine IL-1β is secreted by macrophages and regulates several important processes for the development of endometriotic lesions such as apoptosis, cell proliferation, and differentiation. Furthermore, IL-1β can induce increased expression of VEGF and COX-2, favoring lesion growth through increased vascularization and local estrogen concentration, respectively [88]. Just like the IL-1β, TNF-α that is secreted mainly by peritoneal macrophages in an environment with high estrogen concentrations, induces the production of PGF2α and PGE2 by endometrial epithelial and stromal cells in endometriotic lesions and stromal cells from eutopic endometrium. As a primary effector of inflammatory responses, this cytokine increases the secretion of IL-6, IL-8, and monocyte chemotactic protein-1 (MCP-1) in epithelial endometrial cells [89] and promotes adherence of these cells to the peritoneum. This process plays a crucial role in increasing the invasiveness of endometrial fragments, positively regulating MMPs, and reducing the effect of TIMPs [29].

Elevated serum levels of IL-6 are considered biological markers for the diagnosis of endometriosis. The increased secretion of this cytokine is involved in reducing the cytotoxicity of NK cells, as well as in the activation of macrophages, causing an increase in the secretion of other proinflammatory cytokines, angiogenesis, and cell proliferation [90,91]. Furthermore, this cytokine stimulates the expression of aromatase P450 in endometriotic stromal cells, showing again an interaction between the endocrine system and inflammation [86].

### 2.4. Influence of Cytokines on Infertility

The World Health Organization (WHO) defines infertility as the inability of a couple to establish a clinical pregnancy after 12 months of regular unprotected sex. Among the main causes of infertility, the following stand out: polycystic ovary syndrome, endocrine/anovulatory age, advanced age, and endometriosis, the latter being the most prevalent cause [92]. The mechanisms of infertility related to endometriosis are still not widely understood. However, they may be related to anatomical distortion and tubal occlusion due to pelvic adhesions; impairment of endometrial receptivity; the reduction of folliculogenesis; and oocyte quality mediated by inflammatory cytokines [93]. These molecules are considered important regulatory components, not only for the immune system, but for many other significant functions of the organism [94]. In the female reproductive tract, they can influence embryonic programming, embryo development, reduction of cellular stress and apoptosis, therefore favoring implantation [95]. TNF-α is present in several phases of the process, from follicular rupture to implantation. This cytokine, together with IL-1β, stimulate the production of IL-8 and Granulocyte-Macrophage Colony Stimulating Factor (GM-CSF) that will facilitate ovulation and embryo implantation. Furthermore, TNF-α and INF-γ play an important role in the differentiation/maturation of trophoblastic cells. In the blastocyst stage, the trophoderm (the layer that lines the blastocyst) secretes cytokines that will facilitate embryonic growth, such as IL-10, IL-3, GM-CSF, and IL-6. The latter is involved in the inflammatory reaction during implantation, in uterine receptivity and in the induction of endometrial decidualization [94]. However, the exacerbation of the production of these cytokines in the peritoneal fluid and in the serum of women with endometriosis are harmful to the embryo, directly influencing the interactions between oocyte–sperm, embryonic development, and implantation [96]. Furthermore, high concentrations of these cytokines, which in this scenario are also called embryotoxic cytokines, can induce abortion [97].

The imbalance of cytokine production is also related to the increase in recurrent miscarriages in women with endometriosis and are defined as the occurrence of three or more miscarriages before 20 weeks of pregnancy, where the interruption of the same occurs without any external cause or stimulus [98]. It is known that the production of IL-6 is extremely important in the development and implantation of the embryo; however, in high concentrations, it can limit the interaction of the blastocyst with matrix substrates during the adhesion process, in addition to significantly reducing the sperm motility [96]. Several studies also report that the increase of this cytokine in serum and peritoneal fluid is related to recurrent miscarriage, pre-eclampsia, and preterm births [97,98,99]. Another cytokine that has been linked as a major cause of miscarriage is IFN-γ. Embryos from mice cultured with the serum of women who had recurrent miscarriages did not survive; a fact that was due to the high concentration of IFN-γ in the medium, thus showing the direct relationship of this cytokine with infertility [95].

However, other mechanisms are also involved in endometriosis-related infertility, although most are not fully understood. In addition to the alterations of proinflammatory cytokines in the peritoneal fluid promoting a harmful and pro-oxidative environment and affecting folliculogenesis, oocyte quality and sperm damage, compromised endometrial receptivity and pelvic alterations also influence ovarian function and embryo implantation. In view of the above, all these mechanisms corroborate to the increase in the rate of infertility in women with endometriosis [100,101].

There is increasing evidence of the association between endometriosis and increased risk of pregnancy complications such as placenta previa, preterm birth, hypertensive disorders of pregnancy, small for gestational age, placental abruption, and postpartum hemorrhage [102,103,104,105]. Until now there are several controversies about the effects of endometriosis on perinatal outcomes [106,107]. Additionally, whether prepregnancy treatments such as surgical treatment or hormone therapies for infertility, ovarian endometriosis, and chronic pain due to endometriosis improve perinatal outcomes in subsequent pregnancies remains uncertain and is an important clinical question.

## 3. Diagnosis and Treatment

Currently, the most accurate diagnosis of endometriosis is laparoscopy, with inspection of the abdominal cavity and histological confirmation of suspicious lesions. This type of diagnosis is particularly useful for detecting, through biopsies, occult microscopic lesions in women with and without visible endometriosis. However, this method is expensive and invasive, offering risks to the patient related to the procedure, such as bleeding and infections. Thus, other methods may have advantages when compared to this one, such as imaging methods and biomarkers [108]. Imaging methods such as transvaginal ultrasound (TV-USG) and magnetic resonance imaging (MRI) are suitable for the diagnosis of two phenotypes of endometriosis. The deep infiltrative endometriosis, which can occur in the rectosigmoid, uterosacral, and rectovaginal septum ligaments, and endometriomas, a condition in which USG-TV is the method of first choice. In addition, sigmoid, ileocecal, and urological lesions can be detected with complementary radiological techniques, such as transrectal ultrasound, multidetector computed tomography, or MRI. Scintigraphy can also be used to explore renal function in cases of suspected ureteral endometriosis. Ultrasonography and MRI have high sensitivity (91%) and specificity (98%) to detect and rule out endometriotic lesions, especially deep lesions. However, they are not advisable for the identification of peritoneal lesions, mainly due to the size of the lesions, which is below the detection limit of the devices [17].

Blood, endometrial tissue, and urine biomarkers can be used as markers for the diagnosis of endometriosis. However, such biomarkers are not able to reveal the location of endometriotic lesions. A widely used biomarker is CA-125; despite being found at high levels in endometriosis, CA-125 can be elevated in several diseases. Thus, it has no value as a single test in the diagnosis of the disease [108]. Another option could be the use of questionnaires with internally and externally validated scores of clinical values that could indicate those patients at high-risk of endometriosis [109].

The choice of treatment will depend on the severity of symptoms, the extent and location of the disease, the desire to become pregnant, and the patient’s age. It can be through medication, surgery, or even a combination of both. Pharmacological therapy for endometriosis aims to improve symptoms or prevent recurrence of postsurgical disease [100]. Hormonal treatments act by suppressing fluctuations in gonadotropic and ovarian hormones, resulting in the inhibition of ovulation, menstruation, and a reduction in the inflammatory process [9].

Given this, some drugs create environments such as hyperprogestogenic therapy (combined oral contraceptives and progestins). These drugs are the first choice; they act by inhibiting ovulation, decidualization, and result in a decrease in the size of the lesions. In addition, they are available in a variety of dosage forms, improve pain symptoms in most patients, are well tolerated, and are inexpensive. However, 25% of the patients do not respond to treatment, in addition to having adverse effects such as: sudden bleeding, breast tenderness, nausea, headaches, mood swings, among others [15,110]. Hypoestrogenic therapy (Gonadotropin-Releasing Hormone—GnRH agonists) represents the second line of treatment for this disease. It is an effective drug in the treatment of women who do not respond to combined oral contraceptives or progestins. GnRH agonists provide negative feedback mechanisms in the pituitary, inhibition of gonadotropin secretion, and subsequent downregulation of ovarian steroidogenesis. One of the main disadvantages of these drugs is that they are not administered orally, as they are destroyed in the digestive process, so their use is indicated parenterally, subcutaneously, intramuscularly, via nasal spray, or intravaginally. The use of these drugs is associated with poorly tolerated adverse effects, such as vasomotor symptoms, genital hypotrophy, and mood instability. In addition, GnRH agonists cause a negative calcium balance with an increased risk of osteopenia, although bone loss seems to be reversible if the treatment is limited to a few months [15]. Hyperandrogenic therapy (danazol or gestrinone) produces a pseudomenopause by inhibiting the release of GnRH and the peak of luteinizing hormone (LH) increases the levels of androgen hormones (free testosterone) and decreases estrogen levels (inhibits ovarian production), which causes atrophy of endometriotic implants. However, this class of drugs is not suitable for prolonged treatments, mainly due to androgenic effects, i.e., seborrhea, hypertrichosis, weight gain, unfavorable effects on the distribution of serum lipoprotein cholesterol, a decrease in HDL levels and an increase in LDL levels [111].

Most hormonal treatments for endometriosis focus on inhibiting ovarian estrogen production, rather than blocking estrogen produced locally in endometriotic lesions. Aromatase inhibitors (AI) are a new class of drugs on the market, which are highly specific and act by inhibiting the P450 aromatase enzyme, the final enzyme in the estrogen biosynthesis pathway, in order to stop the local production of this hormone. The use of these drugs significantly reduces the size of lesions, as well as pelvic pain. However, in premenopausal women, AIs need to be combined with other classes of drugs, such as progestin, combined oral contraceptives, or a GnRH agonist. It was observed that the best combination, with minimal adverse effects, was with oral contraceptives or progestin [50]. Its adverse symptoms are loss of calcium in the bones, causing an increased risk of osteoporosis, vaginal dryness, insomnia, vasomotor symptoms, nausea, and headaches. The most potent AIs are those from the 3rd generation, anastrozole and letrozole; they are administered orally and are able to reduce serum levels of 17β-estradiol by 97–99% after one day of use [112]. A randomized controlled clinical study performed by Zhao et al. [113] using a total of 820 patients and aiming to analyze the efficacy and tolerability of letrozole combined with combined oral contraceptives, showed that this combination reduced the intensity of pelvic pain, dyspareunia, and dysmenorrhea throughout treatment, avoiding more severe adverse effects such as bone loss [113].

Nonsteroidal anti-inflammatory drugs are used in association with all the other classes mentioned above. They are widely used in the treatment of chronic inflammatory conditions and are effective in relieving primary dysmenorrhea. However, they only act to minimize symptoms and do not block ovulation. Patients who use these drugs should consider the adverse effects gastric ulcers, cardiovascular events, and acute renal failure [108].

Blocking the specific NFkB DNA-binding sites at promoter regions is another possible strategy [114] that has been used successfully with endometriotic stromal cells in vitro. This inhibition reduces RANTES production and MCP-1 activity induced by IL-1β [115].

Pharmaceuticals with off-target effects on NFkB have also been considered for endometriosis treatment. Thalidomide inhibits NFkB through the suppression of IkB degradation [116]. Treatment of endometriotic stromal cells with thalidomide inhibited TNFa-stimulated IL-8 production and secretion [117]. Thiazolidinediones, a group of drug ligands for PPARg developed for diabetes treatment, was also used as an inhibitor of this pathway and for the reduction of endometriotic stromal cells [118]. A study by McKinnon et al. (2012a) found that the use of these drugs reduced the size of endometriotic lesions in both rats and primates [119]. These drugs, however, also produce adverse effects on skeletal health [119].

Given the upstream convergence of the three MAPK pathways (ERK1/2, p38, and JNK), attempts have been made to target shared upstream mediators. Specific inhibitors of B-raf, Vemurafenib and Dabrafenib, have been approved for use in melanoma; however, significant side effects exist [120]. Similar side effects have also been observed in the use of the MEK inhibitor, Trametinib [121]. Therefore, as long-term therapy is required to treat chronic inflammation, global inhibitors of JNK1 and p38a by orally applied kinase inhibitors at this stage appear unlikely candidates [118].

When symptoms persist or adverse effects outweigh the beneficial effects of drugs, surgical treatment is indicated. Patients who present with anatomical distortion of pelvic structures, adhesions, intestinal or urinary tract obstruction are also eligible to undergo surgery. Conservative surgery consists of cauterization of endometriotic foci and restoration of pelvic anatomy. With the excision of ectopic foci, a significant improvement in pelvic pain and fertility rate are observed, although recurrence of disease symptoms may reappear after surgery. Definitive surgery involves hysterectomy with or without oophorectomy (depending on the age of the patient). This in turn is indicated when there is severity of the disease including persistence of disabling symptoms after conservative drug or surgical therapy. Hysterectomy with bilateral salpingo-oophorectomy with excision of all foci of endometriosis showed cure rates of 90% [17].

### 3.1. New Treatment Options

#### 3.1.1. Elagolix

Elagolix is an orally administered GnRH antagonist capable of partially suppressing estradiol, unlike GnRH agonists, thus preventing the creation of a hypoestrogenic state in patients and reducing adverse effects. This drug proved to be effective in reducing symptoms such as dysmenorrhea and pelvic pain, increasing patients’ productivity at work, and improving their quality of life. Treatment with Elagolix, however, is not recommended for patients who have not responded to treatment with GnRH agonists and antagonists, as well as pregnant women, and patients with osteoporosis and severe liver failure. A randomized double-blind phase three study by Taylor et al. [105] involving 872 women treated with doses of 150 mg and 200 mg of Elagolix for six months showed that almost 70% of these patients reported at least one adverse effect during treatment. Despite this, Elagolix, which has antiproliferative effects, was effective in reducing dysmenorrhea and pelvic pain associated with endometriosis [122].

#### 3.1.2. Resveratrol

It is known that one of the characteristics of endometriosis is exacerbated oxidative stress, which in addition to damaging cells, also influences the expression of NF-кB, associated with the production of cytokines, angiogenic factors, iNOS and COX. Resveratrol, which is a polyphenol, is a new drug that is undergoing clinical trials and can induce the production of antioxidant enzymes, increasing the antioxidant capacity of tissues by 50%, in addition to inhibiting the expression of IL-6, IL-8, TNF-α, and COX-2, presenting an anti-inflammatory and antiangiogenic effect by inhibiting VEGF expression. In addition, this drug also inhibits the production of reactive oxygen species by monocytes, macrophages, and lymphocytes and negatively affects the process of cell proliferation promoted by NF-кB and, by inhibiting it, also reduces the epithelium–mesenchymal transition, essential for the establishment of endometriotic lesions, through the PI3K/AKT/NF-ΚB pathway and the regulation of genes related to this transition [123,124,125].

#### 3.1.3. Curcumin

Some bioactive components found in plants have anti-inflammatory and antioxidant properties that are effective in the treatment of endometriosis, such as curcumin, for example, a natural product extracted from *Curcuma longa* L., traditionally used in some Asian countries to treat diseases. Curcumin can significantly reduce the expression of COX-2, the production of TNF-α and IL-6, and decrease the epithelium–mesenchymal transition, essential for the development of endometriosis [64,108]. In addition, it can also reduce cell proliferation, decrease the size of endometriotic lesions, generate a reduction in matrix metalloproteinase-9 activity, and reduce VEGF expression, thus having an antiangiogenic character [126]. Chowdhury and collaborators demonstrated in their study that the effects of curcumin treatment on eutopic endometrial cells from endometriosis patients significantly reduce the secretion of inflammatory cytokines in these cells, also decreasing the phosphorylation of IKKα/β, NF-κB, STAT3, and JNK signaling pathways [127].

#### 3.1.4. Puerarin

Puerarin is an isoflavonoid found in *Pueraria lobata*, used in the treatment of cardiovascular and neurological diseases. Due to its ability to inhibit aromatase enzyme activity in endometrial cells and suppress cell adhesion and proliferation, it constitutes a possible new treatment against endometriosis [126,128]. A study by Yu et al. [129] showed that treatment with puerarin reduced the levels of estradiol and PGE2 in endometriotic rats, by inhibiting the aromatase enzyme P450 and COX-2, in addition to increasing the expression of 17β-HSD2, thus preventing the growth of endometrial tissue. Kim et al. [130,131,132,133] showed that treatment with an extract obtained from flowers of *P. lobata* reduced the adhesion and migration of these cells, also reducing the formation of endometriotic lesions in mice. Figure 3 summarizes the main treatment options for endometriosis.

Many gynecological societies have published different guidelines in order to help in the diagnosis and treatment of endometriosis. However, the variety of the available treatments and the complexity of this illness leads to significant discrepancies between recommendations. Six national guidelines (the College National des Gynecologues et Obstetriciens Francais, the National German Guideline, the Society of Obstetricians and Gynaecologists of Canada, the American College of Obstetricians and Gynecologists, the American Society for Reproductive Medicine, and the National Institute for Health and Care) and two internationals (the World Endometriosis Society and the European Society of Human Reproduction and Embryology) are widely used around the globe to identify the disease. All the above-mentioned guidelines agree that the combined oral contraceptive pill, progestogens, are therapies recommended for endometriosis-associated pain. Concerning infertility, there is no clear consensus about surgical treatment. Discrepancies are also found in recommendation of the second- and third-line treatments [13].

## 4. Conclusions

Even with several treatments for endometriosis, there is still no cure for this disease. Lesions can survive the treatments with different doses and even resurge after discontinuation of treatment. In view of the limitations of the available treatments, the development of new therapeutic alternatives combining high efficacy with a low incidence of undesirable effects is a continuous goal for research.

## Figures and Tables

**Figure 1 molecules-27-04034-f001:**
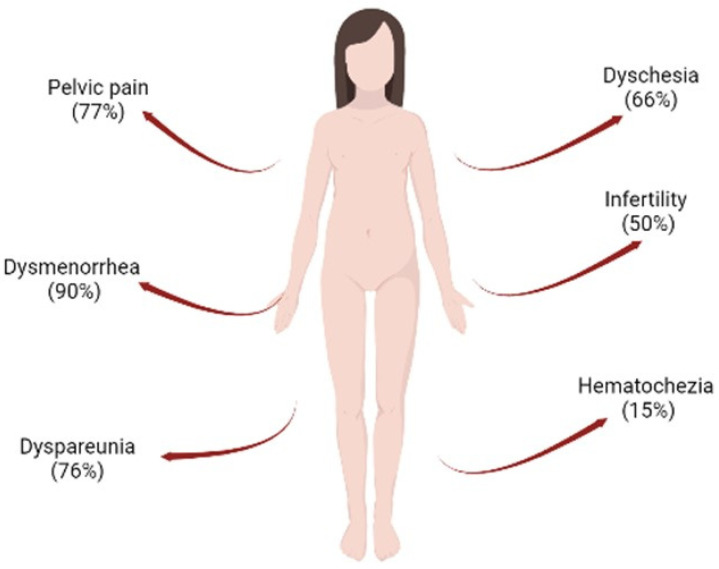
Common associated problems observed in women with endometriosis. The following can be observed: hematochezia (15%), infertility (50%), dyschesia (66%), dyspareunia (76%), pelvic pain (77%), and dysmenorrhea (90%). Figure created in BioRender.com.

**Figure 2 molecules-27-04034-f002:**
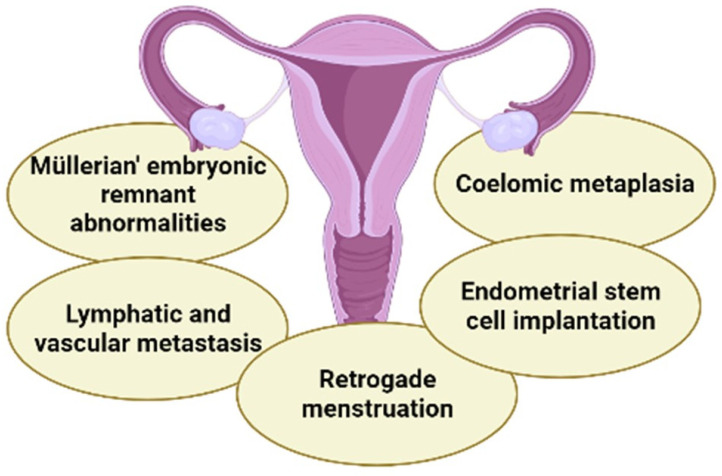
The five main theories explaining endometriosis: Müllerian’ embryonic remnant abnormalities; lymphatic and vascular metastasis; coelomic metaplasia; endometrial stem cell implantation; and retrograde menstruation. Figure created in BioRender.com.

**Figure 3 molecules-27-04034-f003:**
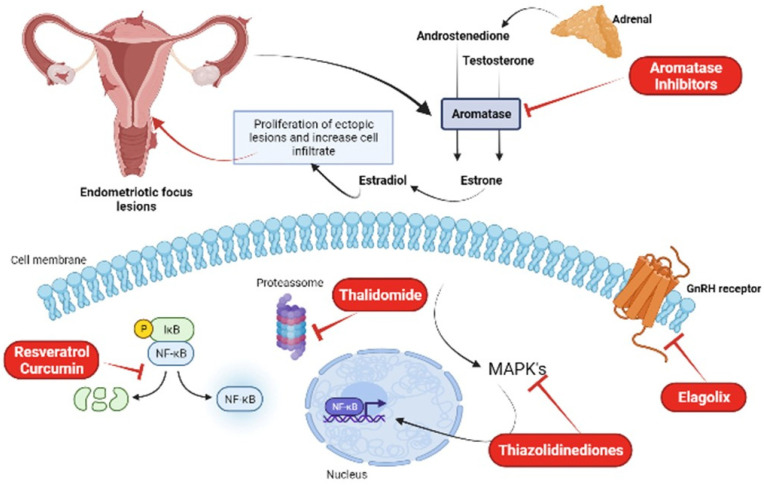
New treatment options for the treatment of endometriosis: aromatase inhibitors, thalidomide, thiazolidinediones, elagolix, resveratrol, and curcumin. Figure created in Biorender.com.

## Data Availability

Not applicable.

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
