# Peer review of "Endometriosis: A Disease with Few Direct Treatment Options"

_molecules, 2022, doi:10.3390/molecules27134034_

Round 1
Reviewer 1 Report
The authors performed a comprehensive review regarding endometriosis. The authors reviewed the characteristics of the disease and available treatments as well as new therapeutic options. This manuscript is well written and the reviewer has only minor comments to improve the manuscript. The reviewer’s comments are listed below.
Introduction
Recently, some systematic reviews have reported that endometriosis is associated with an increased rate of the adverse effects of obstetric outcomes. The reviewer thinks the authors may add descriptions regarding the possible adverse effect of endometriosis on obstetric outcomes.
2.1. Genetic and epigenetic changes
Please cite previous useful studies that examined the different mutation profiles between epithelium and stroma in endometriosis and normal endometrium (Cell Rep. 2018;24:1777-1789.; Hum Reprod. 2019;34:1899-1905.).
If the authors know, with known genes responsible for endometriosis, a novel treatment that blocks endometriosis development may add value to this review.
4.1. New treatment options
A previous study has shown that inflammation leads to the elevation of components of signaling pathways such as mitogen-activated protein kinase (MAPK) in endometriosis and could be a potential target of therapy for endometriosis (Hum Reprod Update. 2016;22:382-403.). If possible, please consider adding this study.
Author Response
Reviewer 1
Comments and Suggestions for Authors
The authors performed a comprehensive review regarding endometriosis. The authors reviewed the characteristics of the disease and available treatments as well as new therapeutic options. This manuscript is well written and the reviewer has only minor comments to improve the manuscript. The reviewer’s comments are listed below.
Introduction
Recently, some systematic reviews have reported that endometriosis is associated with an increased rate of the adverse effects of obstetric outcomes. The reviewer thinks the authors may add descriptions regarding the possible adverse effect of endometriosis on obstetric outcomes.
We added the paragraph:
“There is increasing evidence on the association between endometriosis and increased risk of pregnancy complications such as placenta previa, preterm birth, hypertensive disorders of pregnancy, small for gestational age, placental abruption, and postpartum hemorrhage [12–15]. Until now there are several controversies about the effects of endometriosis on perinatal outcomes [16, 17]. Additionally, whether pre-pregnancy treatments such as surgical treatment or hormone therapies for infertility, ovarian endometriosis, and chronic pain due to endometriosis improve perinatal outcomes in subsequent pregnancies remains uncertain and is an important clinical question.” Please see lines 513-520
2.1. Genetic and epigenetic changes
Please cite previous useful studies that examined the different mutation profiles between epithelium and stroma in endometriosis and normal endometrium (Cell Rep. 2018;24:1777-1789.; Hum Reprod. 2019;34:1899-1905.).
Done. please see lines 232-260
If the authors know, with known genes responsible for endometriosis, a novel treatment that blocks endometriosis development may add value to this review.
To the best of our knowledge there are any treatment focused in genetherapy. It was the reason to not include this area.
4.1. New treatment options
A previous study has shown that inflammation leads to the elevation of components of signaling pathways such as mitogen-activated protein kinase (MAPK) in endometriosis and could be a potential target of therapy for endometriosis (Hum Reprod Update. 2016;22:382-403.). If possible, please consider adding this study.
Some paragraphs were added. Please see lines 614-620

Reviewer 2 Report
The review article by Patricia Ribeiro de Carvalho França et al., „Endometriosis: a disease with few direct treatment options” provides an up-to-date review of relevant research in the field of endometriosis-etipathogenesis, diagnosis and possible treatment. The data presented adds on to similar reviews published on the subject in the last few years.
The following points are recommendations, observations, or questions for the authors:
Introduction is informative, outlining various theories and classification of the disease but lacks a rationale for writing a review on the subject on which many reviews have been published in recent years.
Authors might want to consider citing a recent report by Chapron et al., (https://doi.org/10.1016/j. eclinm.2021.101263) on screening methods for endometriosis diagnosis.
Please remove “uterus” from line 37 and please rephrase lines 39-40.
Line 87: Please remove full stop after sites
Line 126-127: What do authors mean by main criticism of women affected by the disease.
In the section Etiopathogenesis of endometriosis, authors mention oxidative stress, besides other factors, and its involvement in endometriosis will be discussed in detail, however, I did not find any data on the subject.
Line 365: Reference 74 is not relevant to the observation mentioned.
Lines 387-388: Which injury?
Lines 450-454 are vague
Lines 603-604: was effective at both doses in reducing …….. associated with endometriosis.
General comment:
As the Title suggests a review about treatment options, authors did not mention about a recently published, very comprehensive review article “Treatment of endometriosis: a review with comparison of 8 guidelines” by Kalaitzopoulos et al., 2021.
At many places in the manuscript, authors write- these cell type, these hormones and not everywhere it is clear which cell types or hormones authors are referring to.
Author Response
Reviewer 2
Comments and Suggestions for Authors
The review article by Patricia Ribeiro de Carvalho França et al., „Endometriosis: a disease with few direct treatment options” provides an up-to-date review of relevant research in the field of endometriosis-etipathogenesis, diagnosis and possible treatment. The data presented adds on to similar reviews published on the subject in the last few years.
The following points are recommendations, observations, or questions for the authors:
Introduction is informative, outlining various theories and classification of the disease but lacks a rationale for writing a review on the subject on which many reviews have been published in recent years.
Authors might want to consider citing a recent report by Chapron et al., (https://doi.org/10.1016/j.eclinm.2021.101263) on screening methods for endometriosis diagnosis.
We added a phrase. Please see item 4, lines 421-546
Please remove “uterus” from line 37 and please rephrase lines 39-40.
Done
Line 87: Please remove full stop after sites
Done
Line 126-127: What do authors mean by main criticism of women affected by the disease.
In fact the correct should be the complaints of women with the disease. We corrected the phrase. Please see lines 142
In the section Etiopathogenesis of endometriosis, authors mention oxidative stress, besides other factors, and its involvement in endometriosis will be discussed in detail, however, I did not find any data on the subject.
Line 365: Reference 74 is not relevant to the observation mentioned.
corrected
Lines 387-388: Which injury?
We changed injury by implantation of the cysts. Please see lines 405
Lines 450-454 are vague
We agree with the reviewer. We take out this phrase. Please see lines 458-460
Lines 603-604: was effective at both doses in reducing …….. associated with endometriosis.
We corrected the phrase. “at both doses” was take out. Please see lines 669
General comment:
As the Title suggests a review about treatment options, authors did not mention about a recently published, very comprehensive review article “Treatment of endometriosis: a review with comparison of 8 guidelines” by Kalaitzopoulos et al., 2021.
We added a paragraph. Please see lines 689-701
At many places in the manuscript, authors write- these cell type, these hormones and not everywhere it is clear which cell types or hormones authors are referring to.
These errors were corrected

Reviewer 3 Report
The manuscript is well structured and gives a good overview of the latest knowledge of endometriosis characteristics, symptoms etiology diagnosis and treatment. However, the authors also could shortly discuss the findings of the group of group of Tan et al (https://doi.org/10.1101/2021.07.28.453839), showing the coordinated transcriptional programs driving immunotolerance and angiogenesis across eutopic and ectopic endometrium recently uncovered by single cells sequencing approach.
In addition I would suggest that the authors include one or more summary figures visualizing the main take home messages of their review.
To my opinion this slide changes will increase the quality of the manuscript.
Author Response
Reviewer 3
Comments and Suggestions for Authors
The manuscript is well structured and gives a good overview of the latest knowledge of endometriosis characteristics, symptoms etiology diagnosis and treatment. However, the authors also could shortly discuss the findings of the group of group of Tan et al (https://doi.org/10.1101/2021.07.28.453839), showing the coordinated transcriptional programs driving immunotolerance and angiogenesis across eutopic and ectopic endometrium recently uncovered by single cells sequencing approach.
Unfortunately we could not increase the Number of pages and references due to limited space.
In addition I would suggest that the authors include one or more summary figures visualizing the main take home messages of their review.
To my opinion this slide changes will increase the quality of the manuscript.
We added 3 figures along the text.
